# The Emerging Predictive and Prognostic Role of Aggressive-Variant-Associated Tumor Suppressor Genes Across Prostate Cancer Stages

**DOI:** 10.3390/ijms26010318

**Published:** 2025-01-01

**Authors:** Martino Pedrani, Jessica Barizzi, Giuseppe Salfi, Alessandro Nepote, Irene Testi, Sara Merler, Luis Castelo-Branco, Ricardo Pereira Mestre, Fabio Turco, Luigi Tortola, Jean-Philippe Theurillat, Silke Gillessen, Ursula Vogl

**Affiliations:** 1Oncology Institute of Southern Switzerland (IOSI), Ente Ospedaliero Cantonale (EOC), 6500 Bellinzona, Switzerland; martino.pedrani@eoc.ch (M.P.); alessandro.nepote@eoc.ch (A.N.); sara.merler@eoc.ch (S.M.); ricardo.pereiramestre@eoc.ch (R.P.M.); fabio.turco@eoc.ch (F.T.); silke.gillessen@eoc.ch (S.G.); 2Department of Oncology and Hemato-Oncology, Università degli Studi di Milano, 20122 Milan, Italy; 3Istituto Cantonale di Patologia, Ente Ospedaliero Cantonale (EOC), 6600 Locarno, Switzerland; 4Institute of Oncology Research (IOR), 6500 Bellinzona, Switzerland; jean-philippe.theurillat@ior.usi.ch; 5AOU San Luigi Gonzaga, Department of Oncology, University of Torino, 10124 Torino, Italy; 6Medical Oncology Unit, University Hospital of Parma, 43126 Parma, Italy; 7Section of Innovation Biomedicine—Oncology Area, Department of Engineering for Innovation Medicine, University of Verona and Verona University Hospital Trust, 37126 Verona, Italy; 8Faculty of Biomedical Sciences, Università della Svizzera Italiana, 6900 Lugano, Switzerland

**Keywords:** aggressive variant prostate cancer (AVPC), tumor suppressor genes (TSGs), *TP53*, *RB1*, *PTEN*, prostate cancer, prognosis, predictive, targeting

## Abstract

Aggressive variant prostate cancer (AVPC) is characterized by a molecular signature involving combined defects in *TP53*, *RB1*, and/or *PTEN* (AVPC-TSGs), identifiable through immunohistochemistry or genomic analysis. The reported prevalence of AVPC-TSG alterations varies widely, reflecting differences in assay sensitivity, treatment pressure, and disease stage evolution. Although robust clinical evidence is still emerging, the study of AVPC-TSG alterations in prostate cancer (PCa) is promising. Alterations in *TP53*, *RB1*, and *PTEN*, as well as the combined loss of AVPC-TSGs, may have significant implications for prognosis and treatment. These biomarkers might help predict responses to various therapies, including hormonal treatments, cytotoxic agents, radiotherapy, and targeted therapies. Understanding the impact of these molecular alterations in patients with PCa is crucial for personalized management. In this review, we provide a comprehensive overview of the emerging prognostic and predictive roles of AVPC-TSG alterations across PCa stages. Moreover, we discuss the implications of different methods used for detecting AVPC-TSG alterations and summarize factors influencing their prevalence. As our comprehension of the genomic landscape of PCa disease deepens, incorporating genomic profiling into clinical decision making will become increasingly important for improving patient outcomes.

## 1. Introduction

Prostate cancer (PCa) still represents a major health concern affecting men worldwide, although new therapeutic options have arisen [1]. The disease exhibits significant heterogeneity, ranging from indolent localized tumors to aggressive metastatic forms. Clinically, PCa progresses through distinct stages: localized PCa, confined to the prostate and amenable to curative treatments; non-metastatic castration-resistant prostate cancer, characterized by rising PSA despite castrate testosterone levels without detectable metastases; and metastatic disease, which includes hormone-sensitive and castration-resistant phases [2].

Localized PCa is typically managed with active surveillance or active local therapies such as radical prostatectomy (RP) or radiotherapy, often in combination with ADT. Metastatic PCa, however, requires a broader range of interventions, including ADT, androgen receptor pathway inhibitors (ARPIs), chemotherapy, and radiopharmaceutical agents.

Treatment decisions for PCa are primarily based on clinical factors and patient preferences. Currently, international guidelines do not recommend genomic testing for treatment selection in either localized or metastatic-hormone-sensitive settings. Tumor testing for homologous recombination repair gene alterations has only been recommended in the mCRPC setting, where therapies targeting genomic alterations in DNA repair pathways, such as Poly(ADP-ribose) polymerase inhibitors (PARPi), have demonstrated survival benefits.

PCa exhibits significant genomic heterogeneity, and alterations in tumor suppressor genes (TSGs), such as SPOP, *TP53*, *PTEN*, and *RB1*, have emerged as key players in disease development and progression. These TSGs normally function as critical regulators of cellular processes, including AR signaling, cell cycle control, DNA repair, and apoptosis. Inactivation of these genes by various mechanisms, such as mutation or deletion, can disrupt these essential processes and contribute to uncontrolled cell growth and tumorigenesis [3,4].

Several mechanisms, including mutation, deletion, genetic rearrangement, and transcriptional silencing, can affect TSG function during oncogenesis. While *TP53* is often inactivated by mutation, other TSGs may undergo mutation, deletion, or epigenetic silencing. Indeed, interpretation of the literature is challenging, due to variations in reported outcomes, which may arise in part from the diverse assays used to detect TSG alterations and from inter- and intra-patient heterogeneity [5,6].

The advent of comprehensive genomic profiling has provided a growing body of evidence that underscores the influence of TSG alterations in shaping both prognosis and treatment outcomes. Analysis of CRPC tumor samples has revealed a significant enrichment of *TP53*, *RB1*, and *PTEN* alterations in resistant disease, further underscoring their role in driving aggressive disease biology [7,8].

Aggressive variant prostate cancer (AVPC), a subset of prostate cancers that share the clinical, therapy response, and molecular profiles of the small cell prostate carcinomas, has been characterized by a molecular signature of combined tumor suppressor defects in *TP53*, *RB1*, and/or *PTEN* (AVPC-TSGs), determined by either immunohistochemistry (IHC) or genomic analysis [9].

Recent research has highlighted the possible negative prognostic impact of AVPC-TSGs alterations [10] and their association with androgen insensitivity, underscoring the potential for these TSGs to serve as predictive markers for aggressive disease [11]. Conversely, mutations in the SPOP gene may identify a subset of PCa patients with better prognosis and who are particularly dependent on androgen receptor (AR) signaling [12].

While robust evidence to translate these findings into clinical recommendations is still lacking, awareness of the impact of AVPC-TSGs alterations on PCa outcomes has increased over the past decade, as evidenced by discussions at international consensus conferences [13,14].

The Advanced Prostate Cancer Consensus Conference (APCCC) in 2022 acknowledged that tumor genomic profiling does not currently direct treatment decisions for most patients with mHSPC, due to insufficient clinical evidence and the complexity of interpreting diverse assay results for TSGs alterations. However, consensus was reached on the use of triplet therapy for patients with high-volume mHSPC and unfavorable genomic profiles, such as those harboring two or more alterations in *RB1*, *TP53*, and/or *PTEN*. In cases of high-volume mHSPC with a pathogenic SPOP mutation, expert opinion was divided between triplet therapy and ADT + ARPI therapy, underscoring the evolving role of molecular profiling in guiding treatment decisions [13].

The study of AVPC-TSGs alterations in PCa is an evolving field with ongoing research efforts aimed at validating their prognostic and predictive value. In this review, we have summarized the current evidence supporting the significance of the AVPC-associated TSG alterations (*TP53*, *RB1*, *PTEN*) in PCa. As we gain a deeper understanding of the genomic landscape of this disease, the incorporation of genomic profiling into clinical decision making may become increasingly important for personalized treatment strategies and improved patient outcomes.

## 2. Biological Insights and Methodological Approaches to Detect AVPC-Associated TSG Alterations

### 2.1. AVPC-Associated TSG Biology and Impairment

*PTEN*, the phosphatase and tensin homolog located on chromosome 10, plays a critical role in suppressing tumor growth. Functioning as a phosphatase, *PTEN* dampens the phosphatidylinositol 3-kinase (PI3K)-AKT signaling pathway by converting PIP3 to PIP2, thereby regulating cell survival, proliferation, and immune response. Additionally, *PTEN’*s protein phosphatase activity influences cell adhesion and nuclear functions impacting DNA repair and stability. Notably, *PTEN* is the most frequently inactivated tumor suppressor gene in primary PCa, with the vast majority of cases exhibiting *PTEN* loss through genomic deletion [3,15,16].

The retinoblastoma (Rb) tumor suppressor gene, located on chromosome 13q, plays a pivotal role in prostate cancer tumorigenesis. The Rb gene product regulates cell cycle progression by inhibiting the E2F transcription factor, thereby preventing unscheduled entry into mitosis. To stimulate cell proliferation, mitogens and hormones must override the inhibitory effect of Rb by activating cyclin-dependent kinase (CDK) cyclin complexes, which phosphorylate Rb and disrupt its interaction with E2F. This releases the repression of E2F-dependent transcription, allowing the expression of genes essential for cell cycle progression [17,18,19]. In PCa, the failure of *RB1* through mutation or deletion promotes disease progression.

The *TP53* tumor suppressor gene, located on chromosome 17p13, is a critical regulator of cell cycle arrest, apoptosis, and DNA repair, and it is frequently mutated in various human cancers. In PCa specifically, *TP53* inactivation, through mechanisms such as genomic deletion, loss-of-function mutations, or dominant-negative mutations, has been linked to tumor development and progression.

In mCRPC, *RB1* inactivation in combination with *TP53* inactivation can induce a state of cellular plasticity that facilitates the emergence of prostate cancers that have largely lost AR signaling [20]. The latter is further subdivided into a neuroendocrine subtype, a WNT-dependent subtype, and a stem cell-like (SCL) subtype [21].

### 2.2. Determination of AVPC-Associated TSGs Alterations: Comparison of Different Methods

The landscape of TSG alterations in advanced prostate cancer presents a complex interplay between different molecular mechanisms and detection methods. Understanding these alterations requires careful consideration of both the biological mechanisms and the technical approaches used for their detection.

Alterations in *TP53*, *PTEN*, and *RB1* can be detected in clinical setting either by tumor tissue staining or by DNA analysis. Previous studies have analyzed the associations between IHC and genomic sequencing results for *TP53*, *PTEN*, and *RB1* in PCa tissues [22,23,24,25,26]. The partial discrepancies observed between these methods may be partially due to structural variants and promoter methylation, as well as additional factors such as tumor heterogeneity, pre-analytical variables (e.g., sample handling and preservation), technical differences (e.g., sensitivity thresholds and platform variations), biological influences (e.g., post-transcriptional modifications and protein stability), and interpretation challenges (e.g., scoring systems and antibody variability), all of which can impact the detection and interpretation of *TP53*, *PTEN*, and *RB1* alterations in PCa [27,28].

*TP53* alterations in PCa primarily occur through various mutations, with missense mutations being the most common. These mutations can lead to either the accumulation of mutant p53 protein or complete loss of protein expression. Additionally, deletions and other structural variations contribute to *TP53* inactivation [29]. NGS is particularly effective at identifying single-nucleotide variants, showing consistent agreement with loss-of-function transcriptional scores [30]. Detection of *TP53* mutations using NGS or polymerase chain reaction (PCR) has demonstrated high concordance with p53 IHC staining (88–100%) in prostate cancer and other malignancies, such as endometrial and colorectal cancer [23,31,32]. While IHC is highly reliable for detecting missense mutations and provides valuable insights into protein expression patterns, its sensitivity significantly decreases in cases of p53 loss due to nonsense, frameshift, indel mutations, or copy number losses. In such cases, as the overall level of p53 protein is low in the prostate tumor tissue, it is even more difficult to detect p53 protein loss using IHC [22,23,24,25,26]. These limitations suggest that IHC and genomic analysis can provide complementary information in detecting different types of *TP53* alterations. The discordance between these methods becomes particularly significant in mCRPC, likely due to increased heterogeneity of *TP53* alterations in this advanced disease stage [33].

Most prostate tumors with *PTEN* loss inactivate *PTEN* through genomic deletion [3,15,16], though *PTEN* can also be silenced by genetic and epigenetic mechanisms such as genomic rearrangements [34,35,36]. *PTEN* inactivation by mutations (mostly truncating) is relatively rare, occurring in less than 10% of cases. Epigenetic inactivation and hypermethylation of *PTEN* is rarely seen in primary PCa [37]. Other mechanisms that can reduce *PTEN* protein levels or activity include microRNA and noncoding RNA involvement, as well as post-translational modifications like phosphorylation, ubiquitylation, oxidation, acetylation, proteasomal degradation, and subcellular localization [38,39,40].

*PTEN* fluorescence in situ hybridization (FISH) assays, while generally accurate, can be compromised by background signal losses due to the tangential sectioning of nuclei during slide preparation. However, new probe designs have increased the accuracy of these assays [41,42]. IHC protocols validated on the Ventana Benchmark platform in a Clinical Laboratory Improvement Amendments-certified laboratory demonstrated high inter-observer reproducibility and strong correlation of *PTEN* gene loss detected by FISH in comparison to *PTEN* IHC [43,44]. Moreover, there was a high concordance between *PTEN* FISH and IHC in a large radical prostatectomy cohort: 93% of tumors with intact *PTEN* IHC showed no *PTEN* gene deletion, and 66% of cases with *PTEN* protein loss by IHC showed *PTEN* gene deletion by FISH [26]. Indeed, other studies confirmed the correlation of *PTEN* gene loss by FISH and protein loss by IHC in the localized setting [43,45,46,47]. Discordance may arise when *PTEN* loss derives from very small genomic deletions or transcriptional or post-transcriptional regulation, with FISH showing a false negative result. Moreover, IHC may be more sensitive as *PTEN* loss is frequently subclonal and/or focal in primary PCa. On the other side, IHC is not appropriate in detecting hemizygous *PTEN* loss [43,44]. However, whether this discrepancy adds prognostic information remains unclear [48]. Hemizygous *PTEN* loss is more weakly associated with adverse outcomes than homozygous loss [48,49]. Similarly, heterogeneous or partial *PTEN* protein loss is a weaker prognostic indicator than homogeneous or complete loss [43,44].

For the future, circulating tumor cells (CTCs) may be a valid option to determine *PTEN* status, given the high concordance with matched fresh-frozen tissue. However, isolation of CTCs is complex, and technical issues may be easily overcome by using different DNA sources, such as circulating tumor DNA (ctDNA) in blood or in other biological fluids [50,51,52].

*RB1* assessment presents a distinct scenario, as protein loss detection through IHC demonstrates the highest concordance with functional status [30]. This relationship reflects the gene’s tendency toward complete loss of function, making protein-level detection particularly informative. Both hemizygous and homozygous allelic losses correlate with *RB1* protein loss; however, similar to *PTEN*, *RB1* protein expression can be lost even in the absence of allelic loss [9,53,54]. *RB1* can also be functionally impaired through phosphorylation or methylation [55]. Additionally, some tumors with low *RB1* mRNA did not show apparent *RB1* deletions, suggesting potential involvement of focal genomic loss or epigenetic mechanisms in *RB1* inactivation [56].

In a study conducted on 59 tumor samples and 8 patient-derived xenograft (PDX) lines from 42 men with mCRPC meeting AVPC clinical criteria, *RB1* copy number alterations (CNAs) correlated with IHC labeling indices, while no correlation was found for *TP53* [9]. Nevertheless, copy number neutral tumors could still exhibit low *RB1*-positive cells, indicating that CNAs alone are insufficient to detect full pathway alterations [53,54]. This discordance might be even higher in other malignancies. For instance, in a study on small cell lung cancer, among 184 tumor specimens with Rb loss, 29% lacked detectable *RB1* alterations by clinical next-generation sequencing (NGS) pipelines, and 10 cases retained Rb expression despite mutations [57].

In the mCRPC setting, where tumor heterogeneity increases significantly [58], the limitations of individual detection methods become more apparent: IHC may show reduced sensitivity, while DNA-based methods can be affected by tumor content and technical variables. A recent study further elucidated the relationship between IHC, NGS, and transcriptional analysis in the mCRPC scenario. The authors used PDX models to assess *TP53*, *PTEN*, and *RB1* alterations, finding that correlations between IHC and loss-of-function transcriptional scores improved when staining intensity was considered. For *TP53*, single-nucleotide variant (SNV) calls provided the best agreement with the loss-of-function transcriptional score (k = 0.500), compared to weak agreement for IHC results (k = 0.381). For *RB1*, IHC alone had the highest agreement (k = 0.700). For *PTEN*, IHC and CNV calls had similar agreement levels (k = 0.468 and k = 0.490) [30].

These findings indicate that concordance rates between different detection methods can vary significantly across stages. In the mCRPC setting, the heterogeneity of AVPC-TSGs alterations is likely higher, resulting in IHC exhibiting notably lower sensitivity compared to stDNA or ctDNA analysis, as shown in Figure 1.

These findings indicate that concordance rates between different detection methods can vary significantly across stages and types of alterations. IHC sensitivity varies depending on the nature of the genetic alteration: it is particularly effective in detecting p53 missense mutations and complete protein loss, while for *PTEN* and *RB1*, it primarily reflects gene deletions. In the mCRPC setting, the heterogeneity of AVPC-TSG alterations is likely higher, resulting in IHC exhibiting notably lower sensitivity compared to stDNA or ctDNA analysis, as shown in Figure 1. While IHC and NGS can each independently assess AVPC-TSG defects, they offer complementary information based on the type of alteration: NGS excels at detecting both copy number alterations and point mutations, whereas IHC provides direct evidence of protein expression changes. This complementarity may partly explain the discrepancies in the reported AVPC-TSG prevalence, prognostic, and predictive findings, as different assays capture distinct aspects of genomic and protein alterations.

## 3. Prevalence of AVPC-Associated TSG Alterations in Prostate Cancer

Loss of *RB1*, *TP53*, and *PTEN* can occur both early and late in PCa disease, and their prevalence varies across disease stages, with additional complexity introduced by intra-patient molecular heterogeneity [58].

A potential correlation may exist between specific genetic alterations and disease burden. Gilson et al. examined archival primary tumor samples from men with de novo mHSPC enrolled in the STAMPEDE trial. Their findings revealed a higher prevalence of *TP53* mutations in low-volume disease (31%) compared to high-volume disease (21%), while the prevalence of *PTEN*/PI3K pathway alterations did not differ significantly between disease volumes [59]. These observations were corroborated by a separate cohort study from the Memorial Sloan Kettering Cancer Center [60].

Treatment exposure can influence subsequent mutation prevalence, as shown in a subset of patients of the PUNCH trial. *TP53* mutations were enriched in residual tumors of patients treated with neoadjuvant chemohormonal therapy prior to RP for clinically localized high-risk PCa, potentially due to clonal selection [61].

### 3.1. Prevalence of PTEN Alterations in Prostate Cancer

Low rates of *PTEN* loss observed in isolated prostatic intraepithelial neoplasia underscore the relatively late loss of *PTEN* in primary tumors [62,63,64]. In the localized setting, early studies using less sensitive techniques—such as microsatellite analysis—reported a wider range of *PTEN* alterations (10–68%) due to methodological limitations [65,66,67,68]. Specifically, while early Sanger sequencing studies reported high mutation rates in the *PTEN* promoter region, these findings were often confounded by the existence of a *PTEN* pseudogene (*PTENP1*) [69,70]. The actual *PTEN* deletion frequency in this setting may be lower, with subsequent studies reporting *PTEN* deletion in around 15–20% of localized prostate cancer at RP [3,7,15,16,48,58,71]. Moreover, *PTEN* deletions are heterogeneous in up to 40% of primary tumors [43,44,62], posing a challenge for accurate assessment in diagnostic biopsies [72,73,74].

Sequencing studies have revealed that *PTEN* deletion patterns are often conserved between primary and metastatic tumors, indicating that *PTEN* loss frequently arises in the primary tumor before metastasizing [75,76,77]. While these studies suggest that *PTEN* loss is an early event in PCa progression, others have found that the frequency of *PTEN* loss significantly increases in metastatic disease, with rates of 40% in mHSPC [4,71,75,78,79], approaching the frequency observed in mCRPC [59,60,80].

In the mCRPC setting, one study found deep (likely homozygous) *PTEN* deletions in ~30% of patients, with truncating mutations and gene fusions in an additional 10% [7]. Hamid et al., including patients with monoallelic or biallelic TSG inactivation, found a progressively higher frequency of *PTEN* mutation/deletions when comparing localized prostate cancer to mHSPC and mCRPC (16%, 28%, and 67%, respectively) [10].

Interestingly, racial ancestry may influence the frequency of *PTEN* loss, with lower rates observed in African-American men compared to European-American men [65,67,68,81].

### 3.2. Prevalence of RB1 Alterations in Prostate Cancer

Loss of *RB1* is an early event in the development of prostate tumors [82,83,84]. Early studies, limited by small sample sizes and less sensitive techniques (such as global copy number analysis or loss of heterozygosity analysis [84,85,86,87,88,89,90,91,92,93,94,95,96,97]) reported a wide range (18–72%) of 13q deletions, the chromosomal region harboring the *RB1* gene. However, more recent studies employing whole-genome sequencing (WGS) and immunohistochemistry (IHC) have revealed that *RB1* deletions or overexpression of phosphorylated protein are present in at least 25% of PCa patients [52,83,98,99].

In untreated localized prostate cancer, RB expression is lost in 1–20% of patients [15]. WGS of localized prostate cancer revealed a higher incidence of *RB1* deletions in patients treated with neoadjuvant hormonal therapy compared to untreated ones [3,56]. Comparison of primary treatment-naïve and mCRPC samples from the same patient has shown that the main difference, with the exception of AR, was an increase in *RB1* alterations in mCRPC [100], suggesting a role in disease progression.

The frequency of RB loss in mHSPC ranges between 5 and 35% [60,99,101] and between 20 and 60% in patients with heavily treated mCRPC [71] depending on whether only biallelic *RB1* deletion or also monoallelic deletion is considered. In fact, while complete *RB1* loss is rare in mHSPC (compared to approximately 20% of mCRPC), heterozygous loss is common and may approach 30% [99,101,102].

Further complexity is given by temporal and spatial intra- and inter-patient heterogeneity, highlighting the dynamic nature of *RB1* alterations in PCa progression. Rodrigues and colleagues performed *RB1* FISH in 20 matched, same-patient HSPC-CRPC pairs and demonstrated that *RB1* deletion status can change over time with higher frequency in mCRPC (35% versus 65%) [99]. In a large series of mCRPC biopsies, *RB1* IHC revealed heterogeneous *RB1* expression in approximately 28% of cases. Additionally, WGS analysis of CRPC metastases has shown significant intra-patient molecular heterogeneity, with *RB1* expression often varying within the same tumor [100].

These findings suggest that *RB1* loss may be an early event in PCa progression. It can be acquired under the selective pressure of hormonal therapy and during disease progression, contributing to tumor evolution and heterogeneity. It increases in metastatic PCa, in ADT-recurrent PCa, in castration-resistant stages, and particularly in neuroendocrine PCa (NEPC) [103,104], where it is nearly universal (often alongside *TP53* and *PTEN* alterations), compared to primary tumors [7,9,25].

### 3.3. Prevalence of TP53 Alterations in Prostate Cancer

In localized prostate cancer, earlier IHC studies reported a wide range of p53-positive cases (up to 61%), but this may have been due to technical limitations or the inclusion of false positives [105,106,107]. Additionally, alterations of other proteins, such as *PTEN* inactivation, can lead to increased p53 protein expression [108]. Conversely, a different study found 1% p53 positivity in organ-confined (pT2) tumors and 5% in pT3/pT4 tumors [109]. Notably, studies with lower frequencies of p53 alterations (1–20%) often associated them with unfavorable phenotypes and poor prognosis [106,107,110].

*TP53* alterations increase with progressive disease stages, particularly in mHSPC and mCRPC. In mHSPC, alteration rate rises significantly to 25–45%, and in mCRPC, rates can be as high as 73%, as shown by whole-exome sequencing data from 410 mCRPCs where 33% of tumors presented biallelic loss of *TP53* and 32% presented single-copy loss or a pathogenic mutation [3,7,10,15,58,59,60,80,100,101,111,112,113,114,115].

These findings confirm the marked differences in the *TP53* mutation and p53 expression rate across primary, metastatic castration-naïve, and castration-resistant PCa and suggests that *TP53* alterations may be acquired early in disease progression and contribute to the development of lethal PCa.

### 3.4. Prevalence of AVPC-Associated TSG Alterations in NEPC

*RB1* loss is strongly associated with neuroendocrine differentiation (NED) in advanced PCa [116,117,118] and is considered a hallmark of NEPC, where it frequently co-occurs with *PTEN* and *TP53* alterations [9,25]. As shown in mouse models, while the individual inactivation of either *RB1* or *TP53* results in prostatic intraepithelial neoplasia, their combined loss can trigger NED [114]. The loss of *TP53*, *PTEN*, and *RB1* can drive prostate cancer cells to transform into an aggressive, non-AR–driven, neuroendocrine-like phenotype [103,119]. However, *TP53* and *RB1* deficiency, mutation, or deletion alone are necessary but not sufficient for the development of NEPC [114].

A recent meta-analysis including a total of 14 studies with 449 patients concluded that *TP53* is the most frequently mutated gene in NEPC (49.8%). Common CNAs in NEPC included *RB1* loss (58.3%), *TP53* loss (42.8%), and *PTEN* loss (37.0%). *RB1* or *TP53* alterations, as well as their concurrent occurrence, are common in NEPC, with a prevalence of 83.8% for either alteration or 43.9% for both combined. Comparative analyses reveal a higher prevalence of concurrent *RB1*/*TP53* alterations in de novo NEPC compared to treatment-emergent NEPC [120]. Moreover, *RB1* protein loss occurs nearly universally in small cell carcinomas, while only a minority of high-grade primary or metastatic acinar carcinomas with NED harbor *RB1* alterations [25].

### 3.5. Prevalence of Combined AVPC-Associated TSG Alterations in Prostate Cancer

Massively parallel targeted sequencing has revealed differences in the prevalence of combined AVPC-TSGs alteration across PCa stages. In localized prostate cancer, 39% of cases harbored at least one AVPC-TSGs variant, rising to 63% in mHSPC and 92% in mCRPC. The frequency of two or more unique TSGs hits was markedly increased in mCRPC compared with that in mHSPC and localized prostate cancer (73% vs. 28% vs. 11%, respectively), with 50% of mCRPC displaying three hits compared to 0% of localized prostate cancer [10]. In contrast, other studies comparing locoregional and mHSPC tumors reported no statistically significant enrichment in *TP53*, *PTEN*, or *RB1* alterations in the latter disease stage [104].

Discrepancies in reported AVPC-TSG prevalence across studies may be at least partly attributed to whether monoallelic or biallelic inactivation is considered. Notably, Hamid et al. observed a higher frequency of AVPC-TSG alterations compared to other studies that focused solely on biallelic inactivation [3,8]. The reported biallelic loss of two or more TSGs in the study by Hamid et al. was only 0.5% in localized prostate cancer and 4.7% in mHSPC, which is in accordance with previous studies [7,27].

In conclusion, combined alterations in AVPC-TSGs are rare in earlier stages but appear to become prevalent in mCRPC and NEPC, with a notable increase in frequency as the disease progresses. A schematic representation of AVPC-TSGs alteration prevalence across prostate tumor stages, based on data from the main studies included in this review, is illustrated in Figure 2.

## 4. Prognostic Impact of AVPC-Associated TSGs Across Prostate Cancer Stages

AVPC-TSGs such as *TP53*, *PTEN*, and *RB1* play a crucial role in PCa prognosis, with varying impacts across different disease stages. Most studies assessing the impact of AVPC-TSGs on prognosis have been conducted in the localized setting, but more recently, evidence is accumulating also in the metastatic setting.

### 4.1. Localized Prostate Cancer

#### 4.1.1. Loss of Either *PTEN*, *RB1*, or *TP53*

In localized prostate cancer, *PTEN* inactivation correlates with adverse oncological features, including higher Gleason score and extra-prostatic extension, as demonstrated by a meta-analysis of seven studies [121]. Its prognostic value is particularly significant in lower-grade tumors, where *PTEN* status can improve risk stratification and treatment decisions [122,123]. Moreover, among Gleason score 7 or higher tumors, those with *PTEN* loss had a higher recurrence rate compared to those with intact *PTEN* (80% versus 55%) [35]. Importantly, accurate *PTEN* assessment requires analysis of at least two cancer-containing core biopsies [123].

Multiple studies have found a correlation between *PTEN* status and adverse oncological outcomes, including earlier biochemical recurrence, metastasis, and progression to castration-resistant states [35,43,44,48,49,81,124,125,126,127,128,129]. A meta-analysis of 17 studies, encompassing 6744 patients, assessed the impact of *PTEN* deletion on PCa recurrence after RP or brachytherapy [130]. *PTEN* loss was significantly associated with increased risk of biochemical-recurrence-free survival (BRFS) (hazard ratio (HR) 1.79, 95% confidence interval (CI) 1.49–2.16, *p* < 0.001) and recurrence-free survival (RFS) (HR 1.71, 95% CI 1.50–1.95, *p* < 0.001). Further analysis revealed that *PTEN* deletion significantly predicted poor BRFS or RFS in both heterozygous (HR 1.70, 95% CI 1.31–2.21, *p* < 0.001) and homozygous studies (HR 2.54, 95% CI 1.89–3.17, *p* < 0.001). Moreover *PTEN* loss has been associated with a higher risk of death from PCa, independently from racial ancestry [44,81,122,131,132], even when comparing patients with complete *PTEN* loss to those with partial or no loss with the latter serving as the reference group (HR 2.156, 95% CI 1.169–3.976, *p* = 0.014) [133].

*PTEN* loss is 2–5 times more frequent in tumors with ERG gene rearrangement compared to those without [41,43,44,78,134,135,136]. However, the impact of ERG rearrangement and its association with *PTEN* loss remains contentious. Some findings suggest that the presence of *PTEN* loss in the absence of ERG rearrangement associates with the worst outcomes following surgical or conservative therapies in localized PCa [44,132], and larger studies indicate that patients with *PTEN* loss experience poor outcomes—in terms of BRFS—regardless of ERG status [43,49,137]. Notably, a meta-analysis by Liu et al. found no significant association between ERG rearrangement and BRFS, irrespective of *PTEN* status [130].

Loss of *RB1* has been associated with progression to CRPC and metastasis [118,138]. The analysis of 12,427 consecutive RP specimens revealed that a heterozygous deletion of chromosome 13q—observed in 21% of cases—which includes the *RB1* gene, was associated with adverse pathological features including high Gleason grade (*p* < 0.0001), advanced tumor stage (*p* < 0.0001), high preoperative PSA levels (*p* = 0.0125), and early biochemical recurrence (*p* < 0.0001). It was also an independent predictor of prognosis in multivariable (MV) analysis [139]. Notably, this large cohort study focused on the deletion of chromosome 13q, which encompasses a series of other TSGs such as BRCA2, FOXO1, and KLF5, impeding exhaustive conclusions on the specific impact of solely *RB1* deletion.

In a tissue microarray analysis of 2514 RP specimens, positive p53 IHC staining was strongly associated with advanced pT stage (*p* < 0.0001), high Gleason grade (*p* < 0.0001), positive surgical margins (*p* = 0.03), and early biochemical recurrence (*p* < 0.0001) in low- and intermediate-grade localized prostate cancer [109]. At least two more studies found p53 IHC staining results to be associated with BRFS [10,140]. Overexpression of p53 has also been linked to shorter cancer-specific survival (*p* = 0.024) and overall survival (OS) (*p* < 0.05) in patients with L-PC [141,142].

#### 4.1.2. Compound Loss of AVPC-TSGs

In the localized setting, early compound loss of AVPC-TSGs, even mono-allelic, is associated with aggressive disease marked by altered proliferative signaling well before overt castration resistance, with increased risk of recurrence and disease progression [143,144]. Hamid et al. found that patients with AVPC-TSG-altered localized prostate cancer had shorter event-free survival (EFS) (median 2.6 yr, HR 1.95, 95% confidence interval (95% CI) 1.22–3.13) and time to CRPC (median 9.5 mo, HR 3.36, 95% CI 1.01–11.16). Moreover, cumulative gene hits led to an incremental risk of relapse (EFS with no hits as the reference: one gene hit, HR 1.69, 95% CI 0.99–2.87; two to three genes hits, HR 2.70, 95% CI 1.43–5.08). These findings were then validated in an external cohort with similar results. In this study, AVPC-TSG alteration was largely due to mono-allelic deletion, suggesting that hemizygous TSG loss is indeed prognostic [10].

Also, genomic instability and CNA burden have been linked to risk of relapse in localized prostate cancer, with the latter occurring even when adjustments are made for *PTEN* copy number loss, which was independently associated with biochemical relapse [143]. An MV analysis adjusting for mutational and copy number burden did not demonstrate a significant independent association of increasing gene hits and poorer outcomes. Indeed, a direct link between increasing AVPC-TSG loss and relapse or castration resistance has not yet been well established. Nonetheless, cumulative TSG loss may be a marker of broader genomic instability, reflecting increased genome-wide mutations and CNAs. It is also possible that TSG loss itself is a key driver of genomic instability due to its role in genome maintenance and replication fidelity.

### 4.2. Metastatic-Hormone-Sensitive Prostate Cancer

#### 4.2.1. Loss of Either *PTEN*, *RB1*, or *TP53*

The presence of at least one altered AVPC-TSG by genomic sequencing might be a superior predictor of early progression during first-line treatment compared to clinical criteria alone, as shown by one retrospective study [145].

However, emerging data suggest no association between *RB1* alterations and progression-free survival (PFS) or OS in both de novo (HR 0.86, 95% CI 0.39–1.88, *p* = 0.70; HR 0.41, 95% CI 0.09–1.76, *p* = 0.23) and metachronous mHSPC (HR 1.52, 95% CI 0.66–3.55, *p* = 0.33; HR 2.37, 95% CI 0.71–7.98, *p* = 0.16).

Conversely, *PTEN* alterations have been associated with shorter PFS (HR 1.51, 95% CI 1.05–2.18, *p* = 0.03), but not with worse OS [146].

Patients with *TP53*-altered mHSPC have been linked to lower disease-free survival and shorter time to CRPC development [10]. In one study, *TP53* mutations were categorized as loss of function (LOF) versus dominant negative (DN), and their prognostic significance was tested. In univariate (UV) and MV analyses, *TP53* DN mutations showed a statistically significant association with OS for the entire mHSPC population. Additionally, *TP53* mutations (DN and LOF) were associated with worse OS for metachronous mHSPC, while none of these genomic alterations seemed to affect outcomes for de novo mHSPC [146].

#### 4.2.2. Compound Loss of AVPC-TSG Alterations

A multicenter retrospective biomarker study conducted in 218 patients with mHSPC (93 treated with ADT and 125 with ADT + docetaxel) explored the impact of TSG_low_ status (defined when two or more out of the three AVPC-TSGs presented low RNA expression) compared to TSG_wt_. TSG_low_ (19.2%) was independently correlated with shorter OS (HR 2, *p* = 0.002). In the MV analysis, *PTEN*_low_ and *TP53*_low_ correlated with OS (HR 1.5, 95% CI 1.1–2.2, *p* = 0.018; HR 1.6, 95% CI 1.1–2.4, *p* = 0.013; respectively), while no associations were found for low *RB1* [147].

One study found no association between the presence of combined AVPC-TSG alterations and earlier time to CRPC, although evidence suggested inferior OS with increasing AVPC-TSG hits [10]. However, MV analyses adjusting for mutational and copy number burden did not demonstrate a significant independent association of increasing gene hits and poorer survival outcomes. A separate investigation revealed that a combination of *PTEN*/*RB1*/*TP53* mutations was associated with worse OS (HR: 2.67, 95% CI: 1.26–5.69; *p* = 0.01) and PFS (HR: 2.12, 95% CI: 1.15–3.91; *p* = 0.02) in metachronous, but not in de novo mHSPC [146].

### 4.3. Metastatic-Castration-Resistant Prostate Cancer

#### 4.3.1. Loss of Either *PTEN*, *RB1*, or *TP53*

In mCRPC, multiple studies have established an association between *RB1* alterations and poor survival outcomes [100,104,138]. Abida et al. (2019) showed that *RB1* loss was the most significant genomic factor associated with poor clinical outcomes in contemporary cohorts, including shorter survival and reduced time on ARPI treatment (abiraterone or enzalutamide; *p* = 0.002) [104].

In one study, through Cox coefficient analysis of clinical parameters and *TP53* status, three distinct prognostic groups with varying PFS estimates were identified (median 14.7 vs. 7.51 vs. 2.62 months, *p* < 0.0001). These findings were validated in an independent mCRPC cohort initiating first-ARPI (median 14.3 vs. 6.39 vs. 2.23 months, *p* < 0.0001) showing that *TP53* status was superior to any AR perturbation for predicting prognosis [148].

Recently, in patients with mCRPC, *PTEN* loss of function (LOF) has also been associated with decreased survival compared with intact *PTEN* (HR 1.61, 95% CI, 1.07–2.42; *p* = 0.024) [149].

#### 4.3.2. Compound Loss of AVPC-TSG

In a mCRPC patient cohort, a small number of patients (8%) were identified as prognostic outliers, all with wild-type AVPC-TSGs, one of who died after 5.2 yr [10]. The study was not powered to detect a significant difference in OS in the mCRPC cohort. However, there was evidence of an increased risk of death in men with AVPC-TSG-altered tumors (median OS: 4.5 yr, HR 3.26, 95% CI 0.43–24.72, *p* = 0.23) and cumulative gene hits (log rank *p* = 0.13; TSG1 vs. TSG0, HR 5.89, 95% CI 0.69–50.48; TSG2–3 vs. TSG0, HR 2.71, 95% CI 0.35–21.08; Figure 3), though not statistically significant.

## 5. Predictive Role of AVPC-Related TSG

### 5.1. ADT and Androgen Receptor Pathway Inhibitors

In a correlative analysis of the phase II STREAM Trial (ADT and salvage radiotherapy in men with rising PSA after RP), *PTEN* loss signatures were correlated with worse PFS (HR 1.32, 95% CI 1.07–1.64; *p* = 0.01) [150]. In mCRPC, *PTEN* loss by IHC predicted shorter duration of abiraterone treatment (24 vs. 28 wk; HR: 1.6; 95% CI, 1.12–2.28; *p* = 0.009) with a trend towards a decreased PSA 50% response rate (32% versus 43% of patients (*p* = 0.2) [151].

Knockdown or functional inactivation *RB1* in PCa cells prevents androgen-deprivation-induced proliferation arrest and promotes the growth of castration-resistant tumor xenografts [98,152,153]. As early as 1998, it was observed that a higher frequency of *RB1* mRNA downregulation occurred in patients with PCa relapsed after combined androgen blockade therapy, suggesting that *RB1* inactivation contributes to hormone therapy resistance [154]. In 2018, it was demonstrated that neoadjuvant ADT in localized prostate cancer can select for pre-existing subclones harboring oncogenic alterations, including *RB1* loss [56]. Moreover, *RB1* loss is also emerging as a mechanism of resistance in advanced CRPC after treatment with abiraterone and enzalutamide [7,9,25,103].

Wild-type p53 may suppress AR activation with the inability of p53 mutants to down-regulate AR, and several studies have highlighted its significant role in resistance mechanisms [155,156,157]. *TP53* status in the primary tumor could predict a poorer response to subsequent treatment with abiraterone and enzalutamide in mCRPC [158]. As previously mentioned, *TP53* inactivation, particularly biallelic loss, has been associated with significantly shorter progression-free and overall survival in patients with mCRPC treated with abiraterone or enzalutamide, outperforming alterations in the AR or AR-V7 expression [148].

In mHSPC, the presence of at least one AVPC-TSG alteration by genomic tumor sequencing has been associated with lower PFS in patients undergoing treatment with ADT + abiraterone (TSG-alt: 8.0 months vs. TSG-wt 23.2 months) [145].

Combined loss of AVPC-TSGs is associated with increased lineage plasticity, antiandrogen resistance, and lineage switch through upregulation of Sox2, a reprogramming factor notably upregulated in enzalutamide-resistant PCa cell lines [159].

AVPC-TSGs also play a role in maintaining genomic stability, and their loss can lead to increased mutational burden and CNAs. Preclinical work has demonstrated that the combined loss of *TP53* and *RB1* drives AR independence, neuroendocrine differentiation [118,160], lineage plasticity, and resistance to anti-androgen drugs like enzalutamide [117]. Moreover, lineage plasticity can be restored by restoring *TP53* and *RB1* function [118,161], and the knockdown of *RB1*/*TP53* drives the expression of SOX9, promoting resistance to androgen-targeted therapy [161].

These findings were further supported by Nyquist et al., who found that a gene expression signature reflecting *TP53*/*RB1* loss is associated with diminished responses to AR antagonists. Additionally, PCa with *TP53*/*RB1* loss resists multiple cancer therapeutics [114]. Similarly, combined loss of *TP53* and *PTEN* leads to abiraterone resistance [119]. Furthermore, research indicates that the simultaneous loss of BRCA2 and *RB1* in human PCa cells appears sufficient to induce resistance to enzalutamide [162].

### 5.2. Cytotoxic Therapy

In preclinical studies, *PTEN* knockdown models have not shown particular sensitivity to cisplatin or camptothecin, while higher sensitivity to mitomycin C has been described [163].

RB depletion promotes a protective advantage from cytotoxic therapy through AR- and E2F1-driven increase in TNFAIP8, rendering PCa cells less sensitive to TNFα treatment in combination with actinomycin D [164]. Conversely, *RB1*-deleted PCa cells have demonstrated increased sensitivity to DNA damage agents and microtubule interfering agents [98]. Similarly, loss of *RB1* has been associated with sensitivity to cabazitaxel [165].

Mutant p53 may compromise the response of PCa cells to docetaxel. Specifically, knockdown of p53 significantly downregulated p53 phosphorylation and blocked docetaxel-induced apoptotic cell death compared to the vector control [166].

In an mHSPC setting, patients with AVPC-TSGwt tumors had similar PFS when treated with ADT + docetaxel compared to patients with at least one AVPC-TSGs alteration detected by genomic sequencing (AVPC-TSGalt: 7.8 months vs. AVPC-TSGwt: 9.5 months) [145]. However, patients with low RNA expression in two or more AVPC-TSGs derived no benefit in CRPC-free survival and OS from ADT + docetaxel compared to ADT monotherapy. Conversely, a significant benefit was observed in patients with normal AVPC-TSG RNA expression (CRPC-free survival: HR 0.4, *p* < 0.001; OS: HR 0.4, *p* = 0.001) [147].

Notably, combined loss of AVPC-TSGs has been associated with sensitivity to platinum chemotherapy [167]. In 2013, a phase II trial (NCT00514540) of first-line carboplatin and docetaxel and second-line etoposide and cisplatin, including clinically defined AVPC, reported that 74 of 113 patients (65.4%) and 24 of 71 patients (33.8%) were progression free after four cycles in each treatment group, respectively [168]. They attempted to molecularly characterize clinically defined AVPC using PDX models created from tumor tissue taken from patients in the phase II trial. It was found that combined alterations in *RB1*, *TP53*, and/or *PTEN* were more frequent in AVPC than in unselected CRPC from The Cancer Genome Atlas samples [9]. In 2017, they further established an association between AVPC-TSG alterations and sensitivity to platinum chemotherapy [167]. In 2019, the results of a randomized, open-label, phase I–II trial of cabazitaxel plus carboplatin were published. Exploratory analysis revealed that the benefit of adding carboplatin was more evident in patients with AVPC-TSG-altered PCa (defined by either IHC or ctDNA). AVPC-TSG alterations correlated with both longer PFS (HR = 0.35, *p* = 0.00033) and OS (HR = 0.39, *p* = 0.0024) in the combination strategy. Notably, subgroup analysis found no correlation between BRCA2 mutation and PFS or OS (*p* = 0.61 and *p* = 0.35, respectively), further supporting AVPC-TSG alteration as a marker of platinum sensitivity in mCRPC [169]. Recently, a separate study found that patients with PSA progression during cabazitaxel monotherapy could benefit from the addition of carboplatin to cabazitaxel. Notably, in this work, the occurrence of *PTEN*, P53, or *RB1* mutations (present in 26.7% of patients) was not associated with any clinical outcome measure [170].

### 5.3. Radiotherapy

*PTEN* loss may predict an improved response to radiotherapy in patients with PCa. *PTEN* could maintain genomic stability by delaying G2/M-phase progression of damaged cells, allowing time for double-strand break repair by homologous recombination [171]. Similarly, a significant increase in the sensitivity of *PTEN*-deficient cells to radiotherapy has been reported [163]. However, *PTEN* deficiency has also been also associated with radioresistance [172].

RB loss has been shown to increase sensitivity to radiotherapy [98,173]. This is likely due to the role of *RB1* in DNA damage repair, as its loss may impair the cell’s ability to repair radiation-induced damage. *RB1* loss enhanced ionizing-radiation-induced DNA damage through a *TP53*-dependent pathway, while inactivation of *RB1*/*TP53* has been shown to reverse DNA damage and promote radiation survival [174].

A retrospective study of 101 men with de novo low-volume mHSPC—classified according to the presence of high risk (HiRi) mutation including pathogenic mutations in *TP53*, ATM, BRCA1, BRCA2, or *RB1*—investigated the impact of prostate-directed therapy (PDT). Patients with HiRi mutations demonstrated no significant differences in median OS for no PDT versus PDT (*p* = 0.3). Conversely, patients without a HiRi mutation had significant improvement in OS (60 versus 105.3 months, *p* < 0.001) for no PDT versus PDT [175].

### 5.4. PARP-Inhibitors

*PTEN* deficiency in tumor cells leads to impaired DNA repair mechanisms, making them highly sensitive to PARPi in both in vitro and in vivo experiments [115,176,177]. Further evidence has shown that PARP inhibition triggers p53-dependent cellular senescence in a *PTEN*-deficient PCa setting and induces an apoptotic response upon the combined loss of *PTEN* and *TP53*.

The PI3K-AKT signaling pathway limits the efficacy of PARPi monotherapy. Notably, the combination of PARPi with PI3K inhibitors has been shown to effectively synergize to suppress tumorigenesis in human PCa cell lines and in a *PTEN*/Trp53-deficient mouse model of advanced PCa. This combination approach may be promising for overcoming resistance and enhancing tumor suppression in *PTEN*-deficient PCa [178].

*RB1* alterations have been implicated in both the development of resistance to PARPi and an enhanced effect of PARPi when BRCA2 co-loss occurs. PARP inhibition (with olaparib and talazoparib) does not confer an inhibitory effect on *RB1* knockdown cells compared to control cells. However, it causes greater cell growth inhibition in PCa cells that harbor homozygous and heterozygous co-loss of BRCA2 and *RB1* than in BRCA2 knockdown cells [162]. Moreover, loss of *RNASEH2B* confers cancer cell sensitivity to PARP inhibition due to impaired ribonucleotide excision repair and PARP trapping. However, co-deletion of *RB1* and RNASEH2B leads to PARPi resistance in preclinical models. This resistance is mediated, at least partly, by E2F1-induced BRCA2 expression, thereby enhancing HHR capacity. Nevertheless, the loss of *BRCA2* resensitizes *RNASEH2B/RB1* co-deleted cells to PARP inhibition [179].

PCa cells with combined *TP53* and *RB1* loss exhibit high proliferation rates and increased DNA repair activity, leading to resistance to various therapies, including PARPi. Combination of PARP and ATR inhibition has been shown to effectively target *TP53*/*RB1*-deficient PCa, resulting in significant growth inhibition when compared to PARPi monotherapy (*p* < 0.05) [114].

These data suggest that while *RB1* loss alone is not associated with sensitivity to PARPi, the co-loss of BRCA2 and *RB1* increases sensitivity to PARPi in PCa cells compared to BRCA2 loss alone. Conversely, combined *TP53* and *RB1* loss may confer PARPi resistance, which might be overcome by the combination of PARP and ATR inhibition, exploiting the vulnerability of PCa cells with combined *TP53* and *RB1* loss to replication stress. However, clinical validation is still needed, as human data are limited and our understanding of PARPi resistance mechanisms continues to evolve [180].

### 5.5. Lutetium-177 PSMA (LuPSMA) Radioligand Therapy and Radium-223 Therapy

Preclinical research has found AVPC-TSG mutations to be implicated as mediators of radioresistance to LuPSMA [181]. These mutations can cause the emergence of AR-null phenotypes that lack PSMA expression, making the tumor cells less susceptible to targeting by the radioligand [182,183] and leading to poorer outcomes in mCRPC patients.

In a multicenter retrospective analysis of 126 patients with mCRPC treated with luPSMA (46.8% with at least one TSG alteration) after adjusting for relevant confounders, the presence of ≥1 TSGs mutation was associated with shorter PSA PFS (HR 1.93, 95% CI, 1.05–3.54, *p* = 0.034) and OS (HR 2.65, 95% CI, 1.15–6.11, *p* = 0.023) [184]. Two other studies utilizing ctDNA analysis further validated the negative prognostic impact of AVPC-TSGs and PI3K pathway alterations on LuPSMA treatment [185,186]. Crumbaker et al. (2023) using an 85-gene customized sequencing assay from 32 men treated with Lu-PSMA for progressive mCRPC, observed that alterations in *PTEN*, *RB1*, or *TP53* were associated with shorter PSA-PFS (HR 3.4; *p* = 0.0036) and OS (HR 3.3; *p* = 0.0063) [186]. Specifically, *TP53* alterations demonstrated the strongest association with worse PSA-PFS on MV analysis (HR 21.7; *p* = 0.015) [186]. In the study by Vanwelkenhuyzen et al., *TP53* mutations did not correlate with LuPSMA outcomes, while PI3K pathway alterations, predominantly homozygous *PTEN* deletions, were linked to significantly shorter PFS and were identified as an independent predictor of poor prognosis [185]. *RB1* mutations may be associated with lower PFS (6.0 mo compared to 9.0 mo) and OS (13.9 mo versus 26.5 mo) with radium-223 therapy [187].

The illustrations in Figure 3, Figure 4 and Figure 5 collectively highlight the prognostic and predictive roles of *PTEN*, *RB1*, and *TP53* gene alterations, respectively, across prostate cancer stages, emphasizing their influence on treatment sensitivity, resistance mechanisms, and survival outcomes.

## 6. Leveraging AVPC-TSG Alteration Status for Precision-Targeted Therapy Approaches

### 6.1. Single-Agent mTOR Inhibition

The inactivation of *PTEN* results in the activation of the PI3K/AKT/mTOR signaling pathway, which leads to resistance to ADT and poor clinical outcomes. This has prompted the investigation of single-agent mTOR inhibition in clinical settings.

In a phase I study with everolimus (administered at 10 mg daily for 2 weeks) and docetaxel, conducted on patients with chemotherapy-naïve mCRPC, 4 patients had partial metabolic response on FDG-PET, 12 had stable metabolic disease, and 2 had progressive metabolic disease after a 2-week treatment with everolimus alone. Five of the twelve evaluable patients experienced a PSA reduction ≥ 50% during treatment with everolimus together with docetaxel [188].

In a phase II trial of single-agent everolimus that recruited 37 chemotherapy-naïve patients with mCRPC (SAKK 08/08), a confirmed PSA response of ≥50% was observed in two patients (5%), and an additional four patients (11%) experienced a PSA decline of ≥30%. The deletion of *PTEN* was associated with longer progression-free survival (PFS) and better responses [189]. Another phase II trial included heavily pretreated mCRPC patients who were refractory to standard hormonal and chemotherapeutic agents and were administered everolimus at 10 mg daily. In this trial, 32 patients were evaluable for clinical efficacy. No PSA responses were observed, the median PFS was 3.6 months, and the median OS was 10.4 months [190].

Temsirolimus was tested in a phase II trial in men with chemorefractory mCRPC. Median CTC decline was 48%. However, only one patient had a ≥30% PSA decline [191].

### 6.2. Combination Strategies with mTOR Inhibitor

The collective data suggest that single-agent mTOR inhibition has limited clinical utility in men with mCRPC. However, potential strategies to enhance its efficacy have emerged in the preclinical setting. Inhibition of the insulin-like growth factor-1 receptor (IGF-IR) may mitigate the feedback inhibition caused by mTOR inhibitors. This approach could suppress downstream AKT activation, potentially amplifying the antitumor activity of mTOR inhibition. Based on this rationale, combination strategies have been explored in clinical settings.

A phase I study was conducted in mCRPC patients receiving 6 mg/kg cixutumumab (a fully human IgG1 monoclonal antibody, blocks IGF-IR and inhibits downstream signaling) and 25 mg temsirolimus intravenously each week. The results were unsatisfactory, as no patient experienced a greater than 50% decline in prostate-specific antigens (PSAs) from baseline [192].

A phase I study evaluated the combination of oral ridaforolimus, an mTOR inhibitor, with MK-2206, an AKT inhibitor, in patients with advanced solid tumors. Efficacy was specifically evaluated in patients with mCRPC with *PTEN* deficiency. The results were disappointing: no patients with PC responded to the treatment, and only one patient achieved stable disease for at least 6 months [193].

### 6.3. Targeting Both the AR and PI3K/AKT Pathways

In *PTEN*-loss PC models, PI3K/AKT and AR pathways demonstrate reciprocal feedback regulation. Inhibition of one pathway leads to the activation of the other, providing a rationale for co-targeting these pathways. As a result, research efforts in *PTEN*-altered PC have concentrated on simultaneously targeting both the AR and PI3K/AKT pathways to restrict tumor growth and overcome resistance. Among AKT inhibitors, ipatasertib and capivasertib have been the most extensively studied in combination with AR inhibitors.

In the second stage of a phase I, open-label study, conducted in Japanese patients with mCRPC tumors, ipatasertib was administered at a dose of 200 or 400 mg/day in combination with abiraterone and prednisolone in 28-day cycles. Among the six patients were four who had stable disease and one who had complete response showing a safe and tolerable profile of the combination [194].

In a phase Ib/II study conducted in mCRPC, combined blockade with abiraterone and ipatasertib showed superior antitumor activity to abiraterone alone, especially in patients with *PTEN*-loss tumors. Specifically, radiographic PFS (rPFS) was prolonged in the experimental arm, with similar trends in overall survival and time-to-PSA progression. An even larger rPFS benefit was demonstrated in tumors with *PTEN*-loss versus those without. The combination was well tolerated, with no treatment-related deaths [195].

These data suggest that combined AKT and androgen receptor signaling pathway inhibition with ipatasertib and abiraterone is a potential treatment for men with *PTEN*-loss mCRPC [189]. Notably, ipatasertib was associated with increased grade 3/4 adverse events (AEs) and AEs leading to treatment discontinuation compared to placebo. Diarrhea, hyperglycemia, rash, and transaminase elevation were more frequent in ipatasertib-treated patients, appearing rapidly after treatment initiation (median onset: 8–43 days for the ipatasertib arm and 56–104 days for the placebo arm) [190].

In the IPATential150 trial (NCT03072238), a randomized, double-blind, phase III trial, ipatasertib plus abiraterone was compared with placebo plus abiraterone in patients with previously untreated mCRPC with or without *PTEN* loss. Adding ipatasertib to abiraterone and prednisone/prednisolone significantly improved rPFS for patients with mCRPC with *PTEN*-loss tumors by IHC (HR 0.77, 95% CI 0.61–0.98; *p* = 0.034; significant at α = 0.04). Conversely, in the intention-to-treat population, statistical significance at α = 0.01 was not reached (HR 0.84, 95% CI 0.71–0.99; *p* = 0.043). Secondary and exploratory analyses found that ipatasertib improved outcomes (reduced risk of radiographical progression or death) in patients with either *PTEN*-loss or PIK3CA/AKT1/*PTEN*-altered tumors, based on NGS analysis [196]. Notably, ipatasertib was associated with increased grade 3/4 adverse events (AEs) and AEs leading to treatment discontinuation vs. placebo. Diarrhea, hyperglycemia, rash, and transaminase elevation were more frequent in ipatasertib-treated patients, appearing rapidly after treatment initiation (median onset: 8–43 days for the ipatasertib arm and 56–104 days for placebo) [197]. These data suggest that combined AKT and AR pathway inhibition with ipatasertib and abiraterone shows potential for treating *PTEN*-loss mCRPC, but careful management of side effects is necessary.

Capivasertib—a potent, selective inhibitor of AKT1/2/3—was studied in association with abiraterone in an open-label phase Ib study. Twenty-seven patients with mCRPC who had undergone at least one prior line of systemic therapy received abiraterone acetate 1000 mg (orally administered once daily), plus oral prednisone 5 mg (twice daily) with capivasertib 400 mg (orally, twice daily, with an intermittent schedule of 4 days on, 3 days off). Nine participants (33%) showed a 20% or greater PSA decrease during study treatment [198].

In the placebo-controlled randomized phase II proCAID trial, patients received up to ten 21-day cycles of docetaxel and prednisolone and were randomly assigned (1:1) to oral capivasertib (320 mg twice daily, 4 days on/3 days off, from day 2 each cycle), or placebo, until disease progression. Median PFS—the primary endpoint—showed no significant difference between treatment arms (HR 0.92, *p* = 0.32). Median OS, a secondary endpoint, was prolonged in the capivasertib arm (HR 0.54; 95% CI 0.34–0.88; *p* = 0.01). However, this result required prospective validation in future studies. Notably, results were consistent regardless of PI3K/AKT/*PTEN* pathway activation status [199]. Statistical significance was lost at the OS update of the ProCAID trial (*p* = 0.09). In a subsequent OS update of the ProCAID trial, statistical significance was lost (*p* = 0.09). Interestingly, only the subset of patients previously treated with abiraterone and/or enzalutamide appeared to maintain an OS benefit [200].

A phase I clinical trial involved patients with mCRPC who had previously failed treatment with abiraterone and/or enzalutamide. The study administered escalating doses of capivasertib, starting at 320 mg twice daily (b.i.d.), given 4 days on and 3 days off, in combination with enzalutamide at 160 mg daily. Three patients met the criteria for response (defined as PSA decline ≥50%, circulating tumor cell conversion, and/or radiological response), and all of them harbored aberrations in the PI3K/AKT/mTOR pathway [201].

The randomized phase II RE-AKT trial assessed antitumor activity of capivasertib combined with enzalutamide in mCRPC with prior progression on abiraterone and docetaxel. Response rates were similar between enzalutamide/capivasertib and enzalutamide/placebo (19.1% versus 18.8%, *p* = 0.58). In this case, the pre-planned subgroup analyses by *PTEN*_IHC_ status found no differences in the *PTEN*_IHC_ loss subgroup [202].

A recent press release from the CAPItello-281 trial highlighted promising findings in the treatment of *PTEN*-deficient mHSPC. The study, which enrolled 1012 patients with *PTEN*-deficient mHSPC (identified through a centrally tested immunohistochemistry), reported that adding the AKT inhibitor capivasertib to standard-of-care therapy (abiraterone and androgen deprivation therapy) led to statistically significant improvements in radiographic progression-free survival compared to placebo [203]

### 6.4. Other Combination Strategies

Other PI3K/mTOR inhibitors have been studied in association with AR inhibitors.

In a phase I study (NCT02215096), patients with *PTEN*-deficient mCRPC who progressed on prior enzalutamide received once-daily enzalutamide 160 mg plus PI3Kβ inhibitor GSK2636771 at a 300 mg initial dose, with escalation or de-escalation in 100 mg increments, followed by dose expansion. Only one (3%) patient achieved a radiographic partial response (lasting 36 weeks), and four of thirty-four (12%) patients had PSA reductions of ≥50% [204].

Samotolisib (LY3023414), a PI3K/mTOR dual kinase and DNA-dependent protein kinase inhibitor, was tested with enzalutamide in patients with mCRPC following cancer progression on abiraterone. In this double-blind, placebo-controlled phase Ib/II study (NCT02407054), median rPFS was significantly longer in the samotolisib/enzalutamide versus placebo/enzalutamide arm (10.2 vs. 5.5 months; *p* = 0.03, respectively). Among patients with *PTEN* loss, no differences were seen in terms of rPFS (HR 0.66; 95% CI, 0.25–1.72; *p* = 0.40) [205].

The pan-PI3K inhibitor buparlisib (BKM120) was evaluated in a phase Ib study with the dual pan-PI3K/mTOR inhibitor dactolisib (BEZ235) in combination with abiraterone in patients with CRPC. No objective response and few PSA decreases were reported [206].

In a phase II study conducted in mCRPC with rising PSA, an irreversible, pan-isoform inhibitor of class I PI3K, named PX-866, was tested singly and in association with abiraterone. Among 43 patients treated with PX-866 monotherapy, only 2 patients experienced a partial objective responses. No correlation between *PTEN* status and response was seen [207].

### 6.5. Combination Strategies with CDK4/6 Inhibitor

The *RB1* gene alteration status may be a key factor in identifying patients who could benefit from CDK4/6 inhibitor treatments. CDK 4/6 inhibitor blocks proliferation in an RB- and cyclin D-dependent manner in preclinical prostate cancer models, raising the hypothesis that co-targeting AR and cell cycle with CDK 4/6 inhibitor could improve outcomes in patients with mHSPC. CDK4 and CDK6 are known to promote cell cycle progression by phosphorylating the Rb protein.

Notably, the loss of *RB1* has been associated with resistance to CDK4/6 inhibitor monotherapy in various preclinical breast cancer studies. However, it remains unclear as to whether retained Rb expression is necessary for the anti-proliferative effects of CDK4/6 inhibition in PC. Interestingly, several Rb-independent mechanisms of action have been described, such as p107 stabilization and reduced FOXM1 phosphorylation. Despite this, clinical trials investigating CDK4/6 inhibition in PC patients have generally excluded those with Rb deficiency. This approach raises questions about the potential efficacy of these treatments in a broader patient population and highlights the need for further research to fully understand the role of *RB1* status in CDK4/6 inhibitor response in PC.

In a randomized phase II study, the combination of ADT and the CDK 4/6 inhibitor palbociclib in RB-intact mHSPC (determined by tumor IHC) did not impact response rate or PFS [102]. More specifically, ADT monotherapy was compared to ADT + palbociclib. A total of 72 patients with mHSPC underwent metastatic disease biopsy, and 64 had adequate tissue for RB assessment. A total of 62 of 64 (97%) retained RB expression. The PSA undetectable rate at 28 weeks was similar between treatment arms (*p* = 0.5). Radiographic response rate was 89% in both arms [102].

Ribociclib, a CDK4/6 inhibitor, demonstrates preclinical antitumor activity in combination with taxanes. This combination was then evaluated in a phase Ib/II study in mCRPC with progression on ≥1 ARPI. In this study, the authors sought to test the efficacy of ribociclib plus docetaxel in a genomically unselected patient population. No significant differences were detected in median rPFS in patients with baseline *RB1* loss compared with those without. The phase II primary endpoint was 6-month rPFS rate, with an alternative hypothesis of 55% versus 35% historical control. The primary endpoint was met with a 6-month rPFS rate of 65.8% (95% CI: 50.6–85.5%; *p* = 0.005). The presence of *RB1* loss (OR 0.34; 95% CI, 0.0–9.2) was associated with a trend toward a lower likelihood of PSA50 response, although not statistically significant [208].

### 6.6. Future Prospectives

LSD1/KDM1A, an epigenetic factor crucial in CRPC [209], has been shown to be essential for E2F1 function in *RB1*-deficient CRPC. Moreover, Rb inactivation sensitizes CRPC cells to the LSD1 inhibitor treatment in preclinical models [210]. However, no clinical data have yet been published about the activity of the LSD1 inhibitor in PC.

A selection of ongoing clinical trials evaluating emerging targeted treatments based on AVPC-TSG alteration status in patients affected by prostate cancer are summarized in Table 1.

## 7. Conclusions

The present review summarizes the key findings regarding factors influencing the prevalence of AVPC-TSGs and highlights the available evidence on the emerging prognostic and predictive roles of alterations in these TSGs (*TP53*, *PTEN*, and *RB1*) across PCa stages.

The prevalence of AVPC-TSG alterations increases from localized to metastatic disease and is particularly high in NEPC and heavily pretreated mCRPC. Interestingly, low-volume disease has been associated with a higher rate of *TP53* alterations. Exposure to neoadjuvant chemohormonal therapy seems to enrich *TP53* mutations, while neoadjuvant hormonal therapy seems to enrich *RB1* alterations in residual tumors at prostatectomy. Racial ancestry may also influence the frequency of *PTEN* loss, with lower rates observed in African-American men.

Regarding the effect on prognosis, there seems to be differences according to the stage of disease. In localized disease, *PTEN* inactivation and p53 overexpression have been associated with adverse oncological features and shorter BRFS, while the association with *RB1* loss in this setting is less clear. In mHSPC, *PTEN* and *TP53* alterations have been linked to shorter progression-free survival and overall survival, whereas *RB1* alterations have not been consistently linked to survival outcomes. In mCRPC, *TP53*, *RB1*, and *PTEN* alterations are strongly associated with poor survival outcomes, and *TP53* status has been shown to be a superior predictor of prognosis compared to AR perturbations.

Data about the potential predictive role of AVPC-TSG alterations in PCa are accumulating in both the preclinical and clinical setting. *PTEN* loss and *RB1* alterations are associated with reduced effectiveness of ADT combined with an ARPI, particularly in the mCRPC setting. Mutant p53 may compromise the response to docetaxel, and ADT+docetaxel compared to ADT alone might not confer a survival benefit in patients with AVPC-TSG-altered PCa (defined by a transcriptional signature), but these are very preliminary data and need to be validated.

The combined loss of AVPC-TSGs is associated with sensitivity to platinum-based chemotherapy. Patients without alterations in AVPC-TSGs may derive higher benefit from prostate-directed radiotherapy; however, data are limited in this context. *PTEN* deficiency may confer sensitivity to PARPi, while combined *TP53* and *RB1* loss may confer PARPi resistance. Alterations in any of the AVPC-TSGs have been associated with shorter PSA-PFS and OS in patients undergoing luPSMA therapy, potentially indicating a reduced benefit in this subgroup. Additionally, *RB1* deletion may be associated with poor outcomes with radium-223 therapy.

These findings highlight that in the near future, AVPC-TSG assessment might become highly relevant in the clinical decision making for patients with PCa, possibly guiding personalized treatment strategies. The prognostic and predictive clinical implications of IHC and tumor sequencing might differ depending on the PCa stage and the specific gene studied. Knowledge of the limitations and advantages of each methodology used for AVPC-TSG assessment is essential for an accurate and clinically informative evaluation of these gene alterations.

Finally, ongoing efforts are focused on developing targeted treatments based on the status of alterations in AVPC-TSGs. Clinical trials are evaluating targeted approaches based on *RB1* gene status, such as CDK4-6 inhibitors for *RB1*wt tumors, as well as drugs and combinations of drugs targeting the *PTEN*/PI3K/AKT pathway, alongside emerging therapies for *TP53* mutated tumors. Even if these approaches have not yet provided a clinically significant improvement in outcome, they are necessary to aim for personalized treatment strategies, especially for patients with prostate cancer with aggressive behavior to potentially improve their outcomes.

Finally, ongoing efforts are focused on developing targeted treatments based on the status of alterations in AVPC-TSGs. Clinical trials are evaluating targeted approaches based on *RB1* gene status, such as CDK4-6 inhibitors for *RB1*wt tumors, as well as drugs and combinations of drugs targeting the *PTEN*/PI3K/AKT pathway, including capivasertib, which has recently shown potential benefits in radiographic progression-free survival for *PTEN*-deficient mHSPC in the CAPItello-281 trial. Although many of these approaches have not yet demonstrated clinically significant improvements in outcomes, they remain essential for advancing personalized treatment strategies, particularly for patients with aggressive prostate cancer, to potentially improve their prognosis.

## Figures and Tables

**Figure 1 ijms-26-00318-f001:**
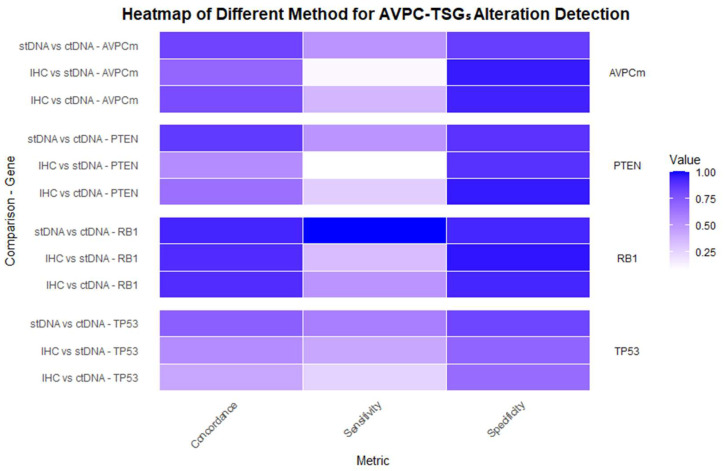
Heatmap of different methods for AVPC-TSG alteration detection in mCRPC. This heatmap visualizes the concordance, sensitivity, and specificity of different detection methods for the genes *TP53*, *RB1*, and *PTEN* in an mCRPC setting. The data used for this analysis are derived from a study published in 2023 [33]. The comparisons include immunohistochemistry (IHC) vs. circulating tumor DNA (ctDNA), IHC vs. solid tumor DNA (stDNA), and stDNA vs. ctDNA. Abbreviations: AVPCm: presence of at least one alteration in *TP53*, *RB1*, or *PTEN* genes; mCRPC: metastatic-castration-resistant prostate cancer; AVPC = aggressive-variant prostate cancer; TSGs = tumor suppressor genes.

**Figure 2 ijms-26-00318-f002:**
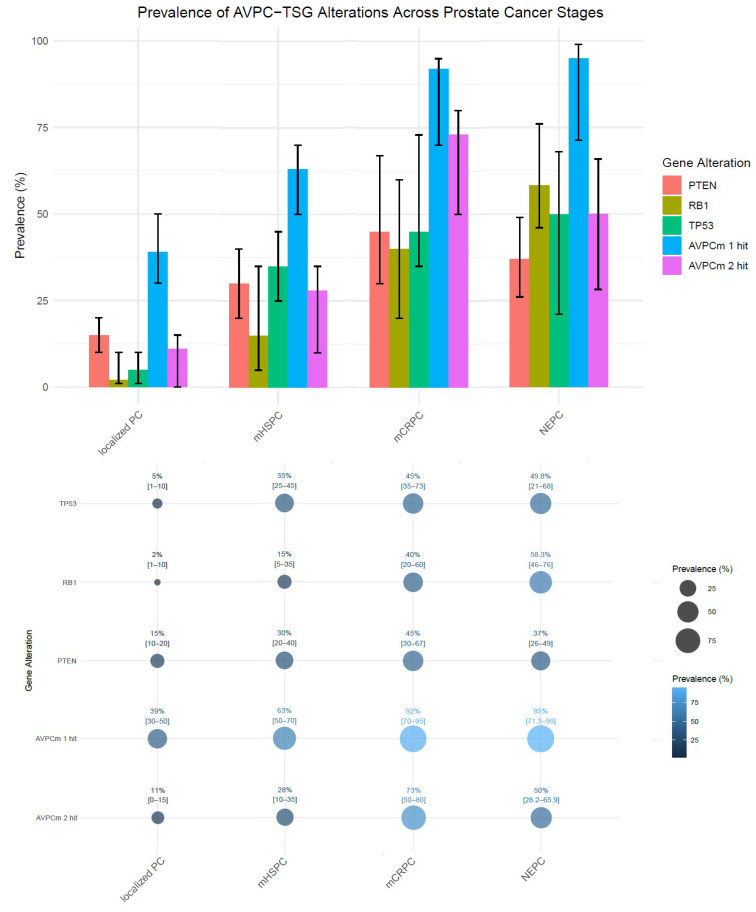
AVPC-TSG alteration prevalence across prostate cancer stages. The figure illustrates the prevalence of aggressive variant prostate-cancer-associated tumor suppressor gene (AVPC-TSG) alterations across different prostate cancer settings. The bars are distinctly colored to represent the different stages of prostate cancer, with error bars indicating extreme values in the literature. Below the main chart, a bubble plot shows the specific prevalence of each genetic alteration in the different stages of prostate cancer, with circles sized proportionally to the prevalence percentage. Abbr: localized prostate cancer (Localized PC); metastatic-hormone-sensitive prostate cancer (mHSPC); metastatic-castration-resistant prostate cancer (mCRPC), neuroendocrine prostate cancer (NEPC); AVPC 1 hit (at least 1 alteration in AVPC-TSGs); AVPC 2 hit (at least 2 alterations in AVPC-TSGs). Note: The data presented are extrapolated from the studies included in the present review. These values are not precise percentages but rather indicative of the range of variation observed across different prostate cancer settings. The higher percentage values of AVPC 1-hit in localized prostate cancer, compared to the sum of single *TP53*, *PTEN*, or *RB1* alterations, might be attributed to the different methods used in studies assessing AVPC 1-hit prevalence.

**Figure 3 ijms-26-00318-f003:**
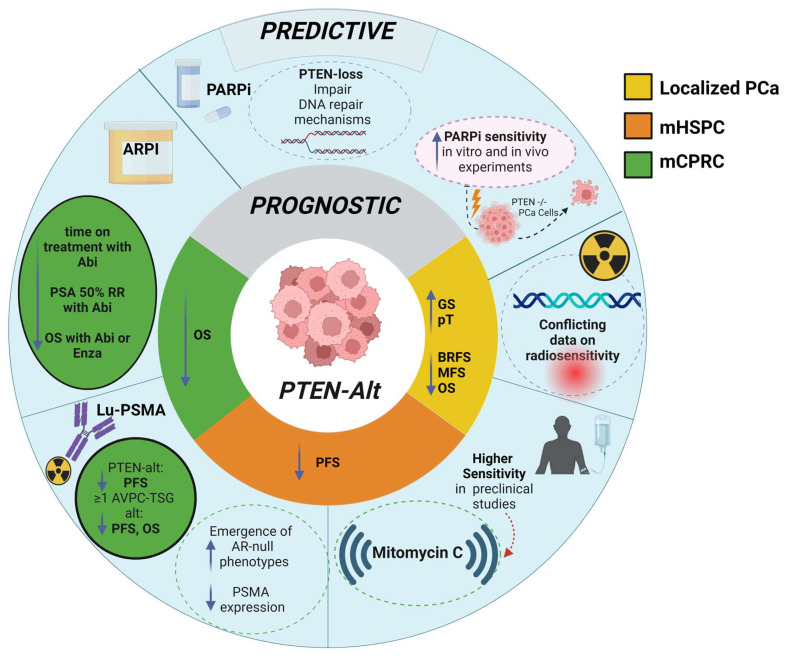
Prognostic and predictive role of *PTEN* alterations in prostate cancer. This diagram illustrates the multifaceted role of PTEN alterations in prostate cancer, highlighting both prognostic and predictive aspects. PTEN loss is shown to impair DNA repair mechanisms, which correlates with increased sensitivity to PARPi in both in vitro and in vivo experiments. The figure also indicates that PTEN alterations can predict the response to ARPI and influence the efficacy of treatments like abiraterone (Abi) and enzalutamide (Enza) in a metastatic castration resistant setting, as evidenced by metrics such as time on treatment and PSA response rates. Prognostically, PTEN alterations are associated with various outcomes such as Gleason score (GS), biochemical-recurrence-free survival (BRFS), metastasis-free survival (MFS), and overall survival (OS). Additionally, the emergence of AR-null phenotypes and changes in PSMA expression are linked to AVPC-TSG status, suggesting a complex interaction with disease progression and treatment response. Conflicting data on radiosensitivity and the impact of treatments like mitomycin C are also noted, underscoring the ongoing need for research in this area. Abbr. PTEN-Alt PCa = PTEN-altered prostate cancer; PARPi = PARP inhibitors; ARPI = androgen receptor pathway inhibitors; Abi = abiraterone; Enza = enzalutamide; Lu-PSMA = lutetium-PSMA; PFS = progression-free survival; AVPC-TSG alt = aggressive variant prostate cancer tumor suppressor gene alterations; OS = overall survival; GS = Gleason score; PT = pathological tumor stage; BRFS = biochemical-recurrence-free survival; MFS = metastasis-free survival; PSA 50% RR = prostate-specific antigen 50% response rate; mHSPC = metastatic-hormone-sensitive prostate cancer; mCRPC = metastatic-castration-resistant prostate cancer; PCa = prostate cancer.

**Figure 4 ijms-26-00318-f004:**
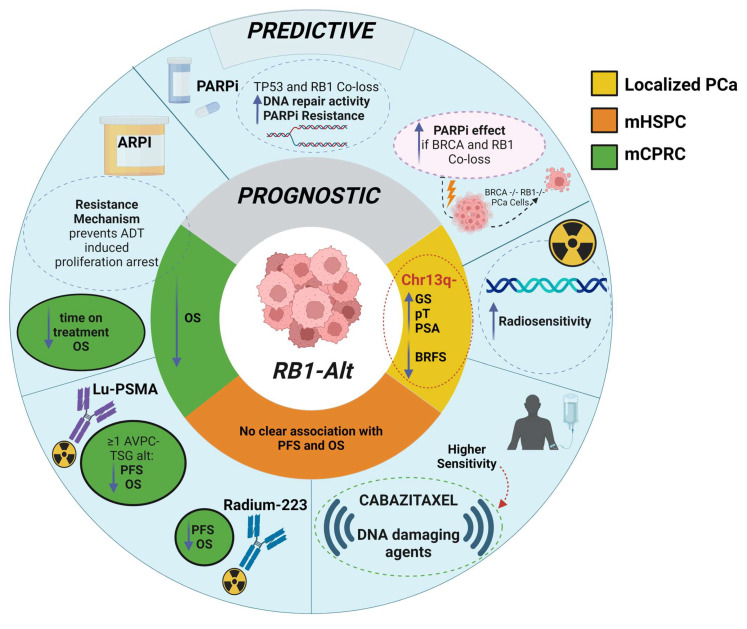
Prognostic and predictive roles of *RB1* alterations in prostate cancer. This figure illustrates the complex role of RB1 alterations in prostate cancer, emphasizing both prognostic and predictive implications. The central diagram highlights RB1-altered prostate cancer (RB1-Alt PCa) and its association with various clinical outcomes and treatment responses. In the localized setting, deletion of the 13q chromosome (which encompasses the RB1 gene) is linked to key metrics such as Gleason score (GS), pathological stage (PT), prostate-specific antigen (PSA) levels, and biochemical-recurrence-free survival (BRFS). RB1 alterations suggest potentially different outcomes depending on treatments: enhanced sensitivity to DNA-damaging agents, cabazitaxel and radium-223, and a potential resistance to PARP inhibitors (PARPi), especially when co-lost with TP53. Conversely, co-alterations in BRCA may enhance the PARPi effect on prostate cancer cells. The figure also notes the impact of RB1 status on radiosensitivity and the effectiveness of ARPI (androgen receptor pathway inhibitors), indicating RB1 as a possible resistance mechanism that prevents androgen-deprivation-induced proliferation arrest. Additionally, it points out that there might be an association between RB1 alterations and progression-free survival (PFS) or overall survival (OS) when considering treatments like Lu-PSMA and radium-223. This comprehensive overview underscores the multifaceted influence of RB1 alterations on the clinical management and therapeutic strategies in prostate cancer. Abbr. RB1-Alt PCa = RB1-altered prostate cancer; PARPi = PARP inhibitors; ARPI = androgen receptor pathway inhibitors; ADT = androgen deprivation therapy; OS = overall survival; Lu-PSMA = lutetium-PSMA; AVPC-TSG alt = Aggressive variant prostate cancer tumor suppressor gene alterations; PFS = progression-free survival; GS = Gleason score; PT = pathological tumor stage; PSA = prostate-specific antigen; BRFS = biochemical-recurrence-free survival; Chr13q = chromosome 13q; mHSPC = metastatic-hormone-sensitive prostate cancer; mCRPC = metastatic-castration-resistant prostate cancer; PCa = prostate cancer.

**Figure 5 ijms-26-00318-f005:**
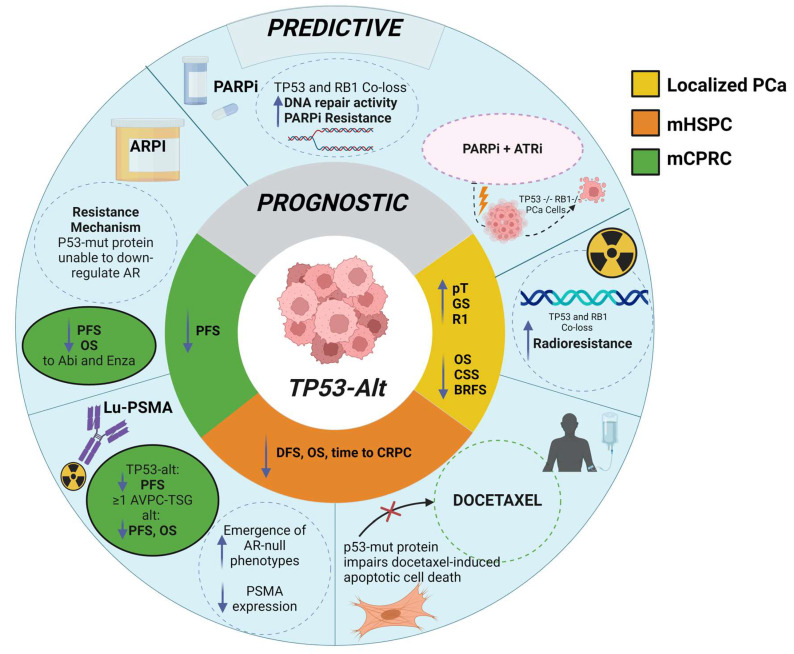
Prognostic and predictive roles of *TP53* alterations in prostate cancer. This figure provides a comprehensive overview of the prognostic and predictive implications of TP53 alterations in prostate cancer. The circular diagram is divided into sections that highlight different aspects of TP53-altered prostate cancer (TP53-Alt PCa). TP53 alterations are associated with several key clinical outcomes such as pathological tumor grade (pT), Gleason score (GS), residual disease post-prostatectomy (R1), overall survival (OS), cancer-specific survival (CSS), and biochemical-recurrence-free survival (BRFS). The diagram indicates that TP53 and RB1 co-loss may contribute to radioresistance in prostate cancer cells. TP53 alterations may predict resistance to PARP inhibitors (PARPi) when there is co-loss with RB1, due to decreased DNA repair activity. The resistance mechanism involves TP53-mutated proteins’ inability to down-regulate androgen receptors (ARs), impacting the effectiveness of AR pathway inhibitors (ARPI). TP53 alterations are linked to decreased progression-free survival (PFS) and overall survival (OS) when treated with agents like abiraterone (Abi) and enzalutamide (Enza) in the metastatic-castration-resistant setting. The emergence of AR-null phenotypes and changes in PSMA expression are noted as significant factors influenced by AVPC-TSG status. TP53-mutant proteins impair docetaxel-induced apoptotic cell death, suggesting a possible resistance mechanism to this chemotherapy agent. Abbr. TP53-Alt PCa = TP53-altered prostate cancer; PARPi = PARP inhibitors; ATRi = ATR inhibitors; ARPI = androgen receptor pathway inhibitors; Lu-PSMA = lutetium-PSMA; PFS = progression-free survival; OS = overall survival; DFS = disease-free survival; AVPC-TSG alt = advanced prostate cancer tumor suppressor gene alterations; GS = Gleason score; PT = pathological tumor stage; R1 = microscopic residual disease; CSS = cancer-specific survival; BRFS = biochemical-recurrence-free survival; mHSPC = metastatic-hormone-sensitive prostate cancer; mCRPC = metastatic-castration-resistant prostate cancer; PSMA = prostate-specific membrane antigen; PCa = prostate cancer.

**Table 1 ijms-26-00318-t001:** Selected ongoing clinical trials evaluating emerging targeted treatments in patients affected by prostate cancer accounting for *TP53*/*PTEN*/*RB1* gene alteration status.

Clinicaltrials.gov Registration Number—Name	Phase	Drug	Type of Drug	Population	Setting	Number of Patients	Endpoints	Current Status	Estimated Primary Completion Date
***TP53*-alt Status**									
NCT06212583	Phase II, randomized, single-center	SOC (hormonal therapy + SABR to the metastatic lesions) vs. SOC + 6-months of niraparib/abiraterone acetate combination pills and prednisone	Hormonal therapy + SABR ± PARPi/ARPI	High-risk metachronous oligometastatic PC with pathogenic mutation in either *TP53, BRCA1/2, PALB2, ATM, BRIP1, CHEK2, FANCA, RAD51B, RAD54L, or MUTYH*	mHSPC	88	18-month PSA, PFS, QoL, safety	Recruiting	2028-12-31
NCT04585750	Phase I/II, open-label, multicenter	PC14586	P53 reactivator	Solid tumors with a *TP53* Y220C mutation	Any	230	ORR, time to response, DOR, PFS, OS, safety, pharmacokinetics and pharmacodynamics	Recruiting	2026-03-17
NCT03903835	Phase III, multi-arm, randomized, cDNA-biomarker-driven	Triplet systemic therapy (ADT + ARPI + docetaxel) vs. SOC	ARPI, chemotherapy	mHSPC arm 2: mHSPC with *TP53*alt and *TMPRSS2-ERG gene* mutations	mHSPC	750, in the whole study	Response rate, PFS, OS, PROMs, cost-effectiveness, safety	Recruiting	2026-12
***PTEN*-alt Status**									
NCT06183736	Phase II, single-arm, open-label, multicenter	CVL237	PI3K p110beta/delta inhibitor	*PTEN*alt advanced solid tumors	Any	98	ORR, DOR, DCR, PFS, OS	Active, not recruiting	2025-06-30
NCT06029998	Phase II, single-arm, open-label, single-center	Bortezomib	Proteasome inhibitor	*PTEN*alt mCRPC	First-line mCRPC or second line after ARPI failure (no other prior systemic treatment)	22	30%PSA response, PSA50, PSA PFS, ORR, DOR, rPFS, PFS, OS, safety	Recruiting	2025-12
NCT05593497	Phase II, single-arm	Abiratorone + capivasertib	ARPI + AKTi	*PTEN*loss high-risk localized PC	Localized, neoadjuvant	30	Pathologic response, surgical complications, safety	Recruiting	2027-05-01
NCT05348577	Phase III, double-blind, randomized, placebo-controlled	Capivasertib + docetaxel vs. placebo + docetaxel	Chemotherapy, AKTi	mCRPC previously treated with ARPI for at least 3 months, preplanned subgroup analysis for *PTEN*loss PC (IHC-evaluated)	mCRPC	1000	OS, rPFS, time to clinical progression	Recruiting; data on the same treatment already available in de novo mHSPC patients (CAPItello-281 study, NCT04493853)	2026-12-21
NCT04586270	Phase I, single-arm, multicenter	TAS0612	RSK, AKT, and S6K inhibitor	mCRPC with documented *PTEN* loss or loss of function mutation	mCRPC	100	DCR, DOR, rPFS, PSA response, safety, pharmacokinetics and pharmacodynamics	Recruiting	2024-08
NCT04060394	Phase I/II, dose escalation and efficacy, randomized, multicenter	LAE001 + prednisone + afuresertib	Androgen synthesis enzyme inhibitor + corticosteroid + AKTi	mCRPC with *PTEN* loss and/or *PIK3CA/AKT/PTEN* alteration who have progressed on, or who are intolerant of 1–3 prior standard treatments	mCRPC	74	rPFS, DOR, DCR, PSA % change, PSA50, time to PSA progression, OS, safety	Active, not recruiting	2023-12-27
NCT03903835	Phase III, multi-arm, randomized, cDNA-biomarker-driven	Capivasertib + docetaxel vs. SOC	AKTi + chemotherapy	mCRPC arm 3: mCRPC with alterations in the PI3K pathway	mCRPC	750, in the whole study	Response rate, PFS, OS, PROMs, cost-effectiveness, safety	Recruiting	2026-12
NCT03218826	Phase I, single-arm, multicenter	AZD8186 + docetaxel	PI3Kbeta inhibitor + chemotherapy	Advanced solid tumors with *PTEN* or *PIK3CB* mutations	Metastatic or surgically unresectable	23	ORR, clinical benefit rate, safety, pharmacokinetics and pharmacodynamics	Active, not recruiting	2022-07-25
***RB1*-alt Status**									
NCT02555189	Phase Ib/II	Ribociclib + enzalutamide	CDK 4/6 inhibitor + ARPI	Chemotherapy-naïve mCRPC that retains RB Expression	mCRPC	46	Determine MTD of ribociclib in combination with enzalutamide; PSA50 at 12 weeks, PSA-PFS and rPFS	Completed results not posted	2023-09-01
NCT05156450	Phase Ib/II	TQB3616 + abiraterone acetate + prednisone	CDK 4/6 inhibitor + ARPI	mCRPC (prior chemotherapy allowed)	mCRPC	30	Evaluate the safety and efficacy of this combined therapy	Unknown status; result not posted	2024-01-01
NCT05268666	Phase I/II	JBI-802	LSD1/HDAC6 inhibitor	Advanced solid tumors	Advanced solid tumors	126	Determine the MTD and recommended phase II dose (RP2D), assess preliminary efficacy	Recruiting	2025-08

AKT = AKR mouse thymoma/protein kinase B; AKTi = AKR mouse thymoma/protein kinase B inhibitor; ARPI = androgen receptor pathway inhibitor; DCR = disease control rate; DOR = duration of response; EFS = event-free survival; IHC = immunohistochemistry; mCRPC = metastatic-castration-resistant prostate cancer; mHSPC = metastatic-hormone-sensitive prostate cancer; ORR = objective response rate; OS = overall survival; PARP-i = poly ADP-ribose polymerase inhibitor; PC = prostate cancer; PFS = progression-free survival; PIK3CA = phosphatidylinositol-4,5-bisphosphate 3-kinase, catalytic subunit alpha; PIK3CB = phosphatidylinositol-4,5-bisphosphate 3-kinase, catalytic subunit beta; PROMs = patient-reported outcome measures; PSA = prostate-specific antigen; PSA50 = percentage of patients achieving 50% PSA reduction; *PTEN*alt = phosphatase and tensin homolog gene alterated; QoL = quality of life; *RB1*alt = retinoblastoma 1 gene altered; rPFS = radiographic progression-free survival; RSK = ribosomal S6 kinase; S6K = S6 kinase; SABR = atereotactic ablative radiation; SOC = standard of care; *TP53*alt = tumor protein P53 gene altered; CDK = cyclin-dependent kinase; MTD = maximum tolerated dose.

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
