# Peer review of "The Emerging Predictive and Prognostic Role of Aggressive-Variant-Associated Tumor Suppressor Genes Across Prostate Cancer Stages"

_ijms, 2025, doi:10.3390/ijms26010318_

Round 1

Reviewer 1 Report

Comments and Suggestions for Authors

Dear Authors,

Thanks for your contribution to this field. This is an interesting state of the art on tumor suppressor genes and prostate cancer and how it is possible to use them for patient benefit. The manuscript is well written and includes updated articles on this subject and data of ongoing work.

Indeed, it will be great if some points are adjusted before its acceptance for publication.

First, the manuscript is presented as a research article, whereas it is a review. Title and paragraphs must be rearranged accordingly.

It will be of interest if the different stages or types of PCa are introduced, which can help in clarifying findings according to the subtype of cancer: metastatic castration-resistant prostate cancer vs. direct metastatic PCa, for example.

Figure 2 is missing some statistical analysis and seems to be repeated throughout the manuscript.

Hoping you can address these points in the manuscript to be accepted for publication.

All the best,

Author Response

Comments Reviewer 1:

Dear Authors,

Thanks for your contribution to this field. This is an interesting state of the art on tumor suppressor genes and prostate cancer and how it is possible to use them for patient benefit. The manuscript is well written and includes updated articles on this subject and data of ongoing work.

Indeed, it will be great if some points are adjusted before its acceptance for publication.

First, the manuscript is presented as a research article, whereas it is a review. Title and paragraphs must be rearranged accordingly.

It will be of interest if the different stages or types of PCa are introduced, which can help in clarifying findings according to the subtype of cancer: metastatic castration-resistant prostate cancer vs. direct metastatic PCa, for example.

Figure 2 is missing some statistical analysis and seems to be repeated throughout the manuscript.

Hoping you can address these points in the manuscript to be accepted for publication.

All the best,

Response to Reviewer 1:

Dear Reviewer,

Thank you for your valuable comments and constructive feedback on our manuscript. We greatly appreciate your insights and suggestions, which have helped us improve the quality and clarity of our work. Below, we address your points in detail:

  1. Manuscript Presentation: We have revised the manuscript to align with its nature as a review article. The title and paragraphs have been rearranged accordingly to reflect this adjustment.
  2. Introduction of Prostate Cancer Stages: We have added a section in the introduction that outlines the differences between various stages and types of prostate cancer, such as metastatic castration-resistant prostate cancer (mCRPC) and direct metastatic prostate cancer. This addition aims to provide better context and clarify findings according to the cancer subtypes.
  3. Figures: The images have been rechecked for accuracy and updated with higher-quality versions. Figure 2 has been revised to ensure it is not repeated unnecessarily throughout the manuscript.

We hope these revisions address your concerns and enhance the manuscript's quality. Thank you again for your thoughtful review and for helping us improve our contribution to this field.

Best regards

Reviewer 2 Report

Comments and Suggestions for Authors

RB1, TP53, and PTEN are the most commonly altered tumor suppressor genes in prostate cancer. This manuscript provided a comprehensive review of these alterations covering their biological relevance, a comparison of different detection methods, their heterogeneity at different disease stages, and predictive and prognostic values. This should be a great piece of reading for anyone who’s interested in pursuing research related to RB1, P53 and PTEN in prostate cancer, and I highly recommend the publication of it. One major missing piece in this review is the discussion of the recently emerging single-cell studies of these alterations in prostate cancer.

Some minor comments:

The figure resolutions are too low, and some of the legends are not legible.

The figure on page 23 is a duplicate.

Author Response

Comments Reviewer 2:

RB1, TP53, and PTEN are the most commonly altered tumor suppressor genes in prostate cancer. This manuscript provided a comprehensive review of these alterations covering their biological relevance, a comparison of different detection methods, their heterogeneity at different disease stages, and predictive and prognostic values. This should be a great piece of reading for anyone who’s interested in pursuing research related to RB1, P53 and PTEN in prostate cancer, and I highly recommend the publication of it. One major missing piece in this review is the discussion of the recently emerging single-cell studies of these alterations in prostate cancer.

Some minor comments:

The figure resolutions are too low, and some of the legends are not legible.

The figure on page 23 is a duplicate

Response to Reviewer 2:

Dear Reviewer,

Thank you for your thoughtful and positive feedback on our manuscript. We greatly appreciate your recognition of the comprehensive nature of our review and your valuable suggestions for improvement.

Regarding your comment on single-cell studies, we fully acknowledge the importance of this emerging field in understanding the heterogeneity and evolution of RB1, TP53, and PTEN alterations in prostate cancer. However, given the clinical focus of our manuscript and space limitations, we were unable to delve into this topic in detail. We agree that single-cell studies provide unprecedented insights into intratumoral heterogeneity, clonal evolution, and spatial distribution of these alterations, and we believe this warrants a dedicated review in the future.

We have also addressed the minor comments:

  1. All figures have been regenerated at high resolution to ensure clarity and legibility.
  2. The duplicate figure on page 23 has been removed, and the numbering has been corrected.

Thank you again for your constructive feedback, which has helped us improve the manuscript. We hope the revised version meets your expectations.

Best regards